# Leaf Anthocyanin Content Retrieval with Partial Least Squares and Gaussian Process Regression from Spectral Reflectance Data

**DOI:** 10.3390/s21093078

**Published:** 2021-04-28

**Authors:** Yingying Li, Jingfeng Huang

**Affiliations:** 1Institute of Applied Remote Sensing and Information Technology, Zhejiang University, Hangzhou 310058, China; lyy0102@zju.edu.cn; 2Key Laboratory of Agricultural Remote Sensing and Information System, Zhejiang Province, China, Zhejiang University, Hangzhou 310058, China; 3Key Laboratory of Environment Remediation and Ecological Health, Ministry of Education, College of Natural Resources and Environmental Science, Zhejiang University, Hangzhou 310058, China

**Keywords:** leaf anthocyanin content, retrieval, partial least squares regression, gaussian process regression

## Abstract

Leaf pigment content retrieval is an essential research field in remote sensing. However, retrieval studies on anthocyanins are quite rare compared to those on chlorophylls and carotenoids. Given the critical physiological significance of anthocyanins, this situation should be improved. In this study, using the reflectance, partial least squares regression (PLSR) and Gaussian process regression (GPR) were sought to retrieve the leaf anthocyanin content. To our knowledge, this is the first time that PLSR and GPR have been employed in such studies. The results showed that, based on the logarithmic transformation of the reflectance (log(1/*R*)) with 564 and 705 nm, the GPR model performed the best (R^2^/RMSE (nmol/cm^2^): 0.93/2.18 in the calibration, and 0.93/2.20 in the validation) of all the investigated methods. The PLSR model involved four wavelengths and achieved relatively low accuracy (R^2^/RMSE (nmol/cm^2^): 0.87/2.88 in calibration, and 0.88/2.89 in validation). GPR apparently outperformed PLSR. The reason was likely that the non-linear property made GPR more effective than the linear PLSR in characterizing the relationship for the absorbance vs. content of anthocyanins. For GPR, selected wavelengths around the green peak and red edge region (one from each) were promising to build simple and accurate two-wavelength models with R^2^ > 0.90.

## 1. Introduction

Anthocyanins exist widely in the plant kingdom, and accumulate in the vacuoles of cells and the tissues of plant vegetative and reproductive organs [1]. They are closely associated with red, blue, and purple coloration of plants [2]. In the leaves, anthocyanins are transiently present in certain developmental stages, such as the juvenile and senescence phases, or persist throughout the leaves’ entire life span [3]. These pigments can be induced by biotic or abiotic stress to help plants resist the adverse effects of the stress, such as herbivory [4], photoinhibition [5], and chilling and freezing [6]. Therefore, anthocyanins are of great significance to plants. The anthocyanin content provides crucial information regarding the plant physiological status.

The traditional method to measure anthocyanin content is wet chemistry (e.g., Amr and Al-Tamimi [7]). This method consists of many procedures, such as field sampling and laboratory chemical assays, which are destructive, and time and labor intensive. However, remote sensing provides a noninvasive and more efficient opportunity to fulfil this task. The leaf reflectance or transmittance can be easily and rapidly measured with a remote sensor in the field. With these spectral data, the anthocyanin content can be retrieved simply and conveniently.

Gamon and Surfus [8] and Sims and Gamon [9] used a red/green reflectance ratio of broad bands to estimate the anthocyanin content. Gitelson et al. [10] built the anthocyanin reflectance index (ARI), and then Gitelson et al. [11] further modified it to construct the mARI (modified ARI). Vina and Gitelson [12] analyzed the sensitivity of four vegetation indices to the anthocyanin content using green reflectance, and found that the visible atmospherically resistant vegetation index (VARI) [13] was highly linearly related to the ratio of anthocyanin content to the sum of the chlorophyll, carotenoid, and anthocyanin content. Unlike these reflectance-based vegetation indices, van den Berg and Perkins [14] built the transmittance-based anthocyanin content index (ACI). Steele et al. [15] modified the ACI index and used the near-infrared and green reflectance instead of the transmittance. Féret et al. [16] developed the radiative transfer model PROSPECT-D, which contains the anthocyanin content as one of the input parameters. The anthocyanin content can be obtained from the inversion of this model. Gitelson and Solovchenko [17] compared the reflectance and absorbance (represented with –log(transmittance))-based approaches, and advised synergistic use to obtain accurate estimation. 

Although these studies have made great progress regarding the retrieval of the anthocyanin content, they are still rare in comparison to the fruitful retrieval studies on leaf chlorophylls and carotenoids. In addition, these studies should also be further examined for applicability under other circumstances. For chlorophylls and carotenoids, many advanced and sophisticated retrieval techniques have been proposed, such as stepwise multiple linear regression [18], partial least squares regression (PLSR) [19,20], continuous wavelet analysis [21], artificial neural network (ANN) [22], and Gaussian process regression (GPR) [23,24]. However, these methods have not been reported to retrieve the leaf anthocyanin content yet. This situation may be caused for two reasons. First, anthocyanins are non-photosynthetic, and are typically present (rich) in certain plant species (e.g., *Tradescantia pallida*), or during certain physiological stages (e.g., senescence) or when plants are under stress. They are generally not as widely abundant in nature as the photosynthetic chlorophylls. Therefore, their retrieval has not been fully addressed by the remote sensing community. Second, the most practical reason is likely the relatively low content of anthocyanins in green leaves, and the consequent difficulty in resolving anthocyanins from other overlapping absorption signals (e.g., chlorophylls), which prevents their retrieval. However, considering the essential physiological significance of anthocyanins for plants, this situation should be improved. More retrieval techniques, especially the advanced ones, should be developed for the leaf anthocyanin content.

Among the advanced retrieval methods, PLSR and GPR are two popular ones. PLSR is a generalization of multiple linear regression. PLSR can analyze data with high collinearity, noise, and numerous independent variables even when the number of independent variables exceeds that of observations [25,26]. PLSR prevails widely in various fields, such as bioinformatics and anthropology. GPR is a machine learning regression algorithm, and it shows good accuracy in retrieval problems, such as estimation of the leaf area index [27], oceanic chlorophylls [28], and soil moisture [29]. This method is nonparametric and based on a Bayesian framework. Nonparametric methods do not require preliminary knowledge, such as the form of a fitting function and, instead, directly infer relationships among the data. To train a GPR model, one only needs to maximize the likelihood function. This process is easy and efficient to realize compared to certain other nonparametric methods. For example, to train an ANN, one may have to do many trials to determine the architecture. Another particularly useful property of GPR is that it also provides the prediction errors [23,30]. With these, it is convenient for an analyst to assess the accuracy of the estimations.

Therefore, in this study, we specifically explored two advanced techniques, i.e., PLSR and GPR, to retrieve the leaf anthocyanin content using spectral reflectance data. To our knowledge, this is the first time that these two techniques have been applied in leaf anthocyanin content retrieval with reflectance. Two aspects were targeted: (1) obtaining a model as simple as possible with only a few wavelengths while maintaining accuracy, and (2) comparing the performance of the various retrieval techniques, particularly the linear PLSR vs. the nonlinear GPR models.

## 2. Materials and Methods

### 2.1. The Overall Process of This Study

Figure 1 summarizes the overall process of this study. First, leaf samples were collected. The spectral reflectance for the leaves was measured, and then the anthocyanin content was determined chemically. Various retrieval methods were investigated to retrieve the leaf anthocyanin content. Their performance was compared and evaluated. The following sections describe the content in detail. 

### 2.2. The Datasets

The leaves of four species, European hazel (*Corylus avellana* L.), Norway maple (*Acer platanoides* L.), and Virginia creeper (*Parthenocissus quinquefolia* (L.) Planch.) in Moscow, Russia, and Siberian dogwood (*Cornus alba* L.) in Nebraska, USA, were sampled from 1992 to 2008. In the spring and autumn, the sunlit leaves of these species contained abundant anthocyanins [31]. Healthy and homogeneously colored leaves without visible damage were selected. The adaxial leaf reflectance was measured with (1) a clip attached to a USB2000 radiometer (Ocean Optics, Dunedin, FL, USA; DOGWOOD2; see the following), and (2) a 150–20 spectrophotometer (Hitachi, Tokyo, Japan) equipped with a 150-mm diameter integrating sphere (DOGWOOD1, HAZEL, MAPLE, and CREEPER). The leaf chlorophyll, carotenoid, and anthocyanin content was analytically determined. The chlorophylls and carotenoids were first quantified, and then the anthocyanins were quantified after extract acidification using concentrated hydrochloric acid. 

The measured data were downloaded from a webpage (see Acknowledgements) and were stored in five Excel tables. All these datasets are mono-species: Siberian dogwood in DOGWOOD1 and 2, European hazel in HAZEL, Norway maple in MAPLE, and Virginia creeper in CREEPER. From these datasets, the records sharing the same spectral region of 436–780 nm were picked and pooled together, forming a total dataset, TOTAL, of 210 samples.

### 2.3. The Basic Thought and Theory on PLSR and GPR

#### 2.3.1. Partial Least Squares Regression 

As there are numerous papers and books specific in the mathematic theory and applications for PLSR (e.g., [25,26,32,33]), only the basic algorithms for this technique are briefly introduced here.

Unlike multiple linear regression, PLSR does not directly use original independent variables (*X*) and responses (*Y*) but utilizes the extracted components (scores) in regression. In PLSR, there are outer relationships:*X = TP′ + E*(1)
*Y = UQ′ + F**(2)
where *T* and *U* are, respectively, the *X* scores and *Y* scores. *P* and *Q* are the loading matrices. The symbol ′ represents the transposition operation of a matrix. *E* and *F** are error matrices. There is an inner relationship:(3)$${\hat{u}}_{h}=b_{h}*t_{h}$$
where ***t_h_*** is the *h*th column vector of *T*, ${\hat{u}}_{h}$ the regression of ***t_h_*** against ***u_h_***, and *b_h_* the regression coefficient. ***u_h_*** is the *h*th column vector of *U.* The mixed relationship is
*Y = TBQ′+ F*(4)
where *B* is the coefficient matrix and *F* is the error matrix. The Euclidian or Frobenius norm of *F* is to be minimized.

When determining the number of components (*a*) for the model, as many components as the rank of *X* can be extracted. However, not all components are necessary, considering the overfitting and that components of a higher order typically describe noise. Generally, *a* can be determined through cross-validation to test the predictive significance of the component. First, the data are divided into *g* (e.g., 10) groups. Second, with the *a* component(s), PLSR models are built on reduced data with one group deleted (*g* models in total). From each model, the differences between the actual and validated responses are calculated. The squares of all these differences are summed together to form the predictive residual sum of squares (PRESS). Similarly, with all data used, one PLSR model is built, and the corresponding “sum” is obtained to form the sum of squares (SS). Third, the ratio PRESS*_a_* /SS*_a_*_−1_ is calculated. If this ratio is smaller to around 0.9 for at least one of the *Y*-variables, adding the *a*th component is regarded to be significant to reduce the error in prediction, and vice versa.

In our study, PLSR was performed in MATLAB (Version R2017b; MathWorks) with the function *plsregress*. When determining *a*, the TOTAL dataset was randomly evenly divided into 10 groups, and the threshold for the component significance was set at 0.95^2^.

#### 2.3.2. Gaussian Process Regression

This section outlines the primary algorithm for GPR according to [34]. Let us consider a dataset 𝒟 = {(***s_i_***, *z_i_*) | *i* = 1, 2, …, *n*} = (*S*, ***z***), where ***s_i_*** ∈ ℝ*^b^* is an observation (e.g., a spectrum) with *b* variables (e.g., wavelengths), and the scalar *z_i_* the response (e.g., anthocyanin content) to ***s_i_***. For brevity, 𝒟 is further aggregated with *S* = [***s*_1_**_,_
***s*_2_***_,_* …, ***s_n_***] and ***z*** = [*z*_1_, *z*_2_, …, *z_n_*]*’*. Under the Gaussian process (GP) framework, *z* is the sum of a latent function *f* (∙) and an additive independent noise *ε*, *z* = *f* (***s***) + *ε*, where
*f*(***s***) ~ 𝒢𝒫(**0**, *k*(***s***, ***s*^#^**)) (5)
(6)$$\varepsilon ∼𝒩(0,\;\sigma_{n}^{2})$$
***s*** and *z* represents any one observation and the corresponding response, respectively. Equation (5) means a GP is assumed on the latent function *f*(∙). A GP is defined as a collection of random variables, any finite number of which have a joint Gaussian distribution, and this is specified by a zero mean function and a covariance (kernel) function *k*(***s***, ***s*^#^**). Equation (6) assumes the noise *ε* to follow a Gaussian distribution with a zero mean and $\sigma_{n}^{2}$ covariance.

For the test data points (*S_*_*, ***z_*_***) with *n_*_* observations, the prior joint distribution of ***z*** and ***z*_*_** under the priors Equations (5) and (6) is
(7)$$\left[\begin{array}{c} z\\ z_{*} \end{array}\right]∼\;𝒩(0,\;\left[\begin{array}{cc} K\left(S,\;S\right){+\sigma }_{n}^{2}I & {K(S,\;S}_{*})\\ {K(S}_{*},\;S) & {K(S}_{*}{,\;S}_{*}) \end{array}\right])$$
where *K*(*S*, *S_*_*) is an *n* × *n_*_* covariance matrix evaluated at all pairs of training and test observations using *k*(***s***, ***s*^#^**), and similarly for *K*(*S*, *S*), *K*(*S_*_*, *S*), and *K*(*S_*_*, *S_*_*). *I* is an identity matrix. The posterior distribution for ***z*_*_** is
***z*_*_** |*S*, ***z***, *S_*_* ~ N(***μ_*_***, *Σ_*_*)(8)
where
(9)$${\mu_{*}}_{\;}=K(S_{*},\;S)[K(S,\;S)+\sigma_{n}^{2}I{]}^{-1}z$$
(10)$${\Sigma_{*}}_{\;}=K(S_{*},\;S_{*})-K(S_{*},\;S)[K(S,\;S)+\sigma_{n}^{2}I{]}^{-1}\;K(S,\;S_{*})$$
***μ*_*_** and *Σ_*_* are, respectively, the mean and covariance for the prediction at *S_*_*. In other words, ***μ*_*_** is the best predicted value, and *Σ_*_* shows a confidence measure to this prediction. The superscript −1 represents calculating an inverse matrix.

The above is the basic thought for GPR. Apparently, the covariance function *k*(***s***, ***s^#^***) plays a core role. In GPR, various covariance functions can be used. A typical choice is the automatic relevance determination (ARD) [35] squared exponential:(11)$$k(s,s^{\#})=\sigma_{f}^{2}\exp[-\frac{1}{2}\sum_{m=1}^{b}\frac{{\left(s_{m}-s_{m}^{\#}\right)}^{2}}{\sigma_{m}^{2}}]$$
where *σ_f_* is the signal standard deviation, and *σ_m_* is the characteristic length scale for each variable. Both hyper-parameters are > 0. The reciprocal of *σ_m_* represents the relevance of each variable, and a low value of *σ_m_* indicates a high informative content of the variable in regression.

To train the GPR model, we use the marginal likelihood
(12)p(z|S)=∫p(z|f,S)p(f|S)df
with the marginalization over the function values ***f***. The log marginal likelihood function is
(13)$$\log[p(z|S)]=-\frac{1}{2}z^{\prime }{\left[K(S,S)+\sigma_{n}^{2}I\right]}^{-1}z-\frac{1}{2}\log\left|K(S,S)+\sigma_{n}^{2}I\right|-\frac{n}{2}\log2\pi$$

To maximize the marginal likelihood, the partial derivatives of Equation (13) toward the hyper-parameters ***θ*** (*σ_f_*, *σ_m_*, *σ_n_*) are calculated, where a gradient-based optimizer is applied. Thus, the optimal solution for the hyper-parameters is determined. Then, with the optimal hyper-parameters, the prediction at *S_*_* can be calculated through Equations (9) and (10).

In our study, GPR was run using the public GPML code (Version 4.2; see Acknowledgements) in the MATLAB (Version R2017b; MathWorks) environment.

### 2.4. Wavelength Selection, Model Building for PLSR and GPR

TOTAL had 345 wavelengths. Both PLSR and GPR can build a model with all of these wavelengths. Such a model is clearly too complicated (and likely overfitting) to be interpreted and transferred to other datasets. Therefore, some wavelengths must be eliminated to simplify the model.

With this in mind, the first problem is how to determine to retain or remove a variable in model building. Thus, an indicator should be introduced to assess the importance of the variables. For GPR, the hyper-parameter *σ_m_* indicates such an importance assessment. A large *σ_m_* value indicates low importance, and vice versa [36]. For PLSR, we used the regression coefficients (*β*). A large absolute value of *β* (|*β*|) indicates high importance [37].

To build a regression model as simple as possible, first, the TOTAL dataset was randomly evenly divided into 10 subsets. These 10 subsets were also used in all the model calibrations and validations below. Second, the sequential backward band removal (SBBR) algorithm [38] was applied. This algorithm consists of two steps. (a) With all available wavelengths (345 at the beginning), a model was calibrated on nine subsets. The remaining subset was used for model validation. During calibration, the wavelength importance indicator and indicator rankings were recorded. The indicator and indicator rankings in the ten times of calibration were, respectively, added together. Using either the sum of the indicator or the sum of the indicator rankings to evaluate the wavelength importance, the least important wavelength for retrieval was removed. (b) With the remaining wavelengths, step (a) was repeated until only one wavelength was left in the model. This last wavelength should be the most important one associated with the response in retrieval. Third, starting from this last wavelength, several mutually distant (distance > 10 nm) wavelengths were picked out from the sequential regression models. These wavelengths consisted in the final regression models. A distance was specified to avoid a strong correlation between too near wavelengths. The 10 nm was selected because this is the lowest spectral resolution a hyperspectral instrument should have.

In addition to the reflectance (*R*), the logarithmically transformed data log(1/*R*) were also used, because this transformation can be regarded as a measure of leaf absorbance [18].

### 2.5. Other Retrieval Methods

We also examined some of the other retrieval methods, mentioned in Section 1. These methods are summarized in Table 1. The *R_x_* in the five vegetation indices represents the reflectance at the wavelength of *x* nm or in a spectral range (e.g., red). For the red/green ratio index, two forms were used. Red/Green-1 (specified as *R*_675_/*R*_550_) is a narrow band ratio at single wavelengths, while Red/Green-2 is a broad ratio. For mARI, the first wavelength (530–570 nm) had the strongest correlation with the anthocyanin absorption, and was determined at 549 nm. Then, the second (690–710 nm) and third (NIR, 760–780 nm) wavelengths were optimized (to make the R^2^ of linear regression for mARI vs. the anthocyanin content the largest) at 699 and 760 nm, respectively. Thus, mARI was determined as (1/*R*_549_ − 1/*R*_699_) × *R*_760_. mACI was specified as *R*_780_/*R*_550_.

### 2.6. Model Calibration, Validation, and Evaluation

For the vegetation indices in Table 1 and the final determined PLSR and GPR models, 10-fold cross-validation was performed on the TOTAL dataset. During each calibration-validation process (including those in Section 2.4), the predicted and the actual anthocyanin content was fitted with a linear model at a confidence level of 95%. The R^2^ and root-mean-square error (RMSE) were calculated. The resulting 10 RMSEs and 10 R^2^ values were, respectively, averaged as accuracy estimators. In addition, for the final PLSR and GPR models, the correlation coefficient (*r*) and the scatterplots for the measured vs. predicted anthocyanin contents were calculated and drawn, respectively. As for PROSPECT-D, the prediction was directly derived with the model inversion.

## 3. Results

### 3.1. Statistics for the Leaf Pigment Content

Table 2 shows the statistics for the datasets. The chlorophyll content had the largest range and standard deviation, while those for the carotenoids were the smallest. Figure 2 displays the correlation coefficients between the pigments of the TOTAL dataset. The chlorophylls and carotenoids had a relatively strong correlation, while anthocyanins had a weak correlation with both chlorophylls and carotenoids.

### 3.2. Retrieval with GPR

Figure 3 shows the retrieval results with GPR. Generally, the accuracy was quite stable with three or more wavelengths in the models. There are several abrupt fluctuations around some numbers of wavelengths, but this did not influence the general trend of the accuracy. With a given number of wavelengths (≥3, except those fluctuations), the retrieval between calibration and validation was not much different. For example, at 120 wavelengths, based on *R* and using the sum of the *σ_m_* rankings as the wavelength importance indicator, the R^2^ values were both 0.93, and the RMSE values were 2.15 and 2.24 nmol/cm^2^ in calibration and validation, respectively. Moreover, the retrieval based on log(1/*R*) was better than that based on *R.* To assure high accuracy of the final models, the retrieval based on log(1/*R*) was further investigated.

Table 3 lists the retrieval results of the last five models with SBBR. From iterations #1–5 with strategies 1 and 2, two optimal wavelengths were selected in sequence to build the final GPR models. The retrieval results with these two final models are listed in Table 4. These two final models resulted in nearly the same accuracy, and the model with strategy 1 was slightly better than that with strategy 2. With only two wavelengths, both models maintained the similar accuracy as the models with ≥3 wavelengths in Table 3. Figure 4 shows the scatterplots for the actual vs. the predicted content from the two final models in validation. The dots are uniformly distributed around the 1:1 line, indicating no apparent bias in prediction.

### 3.3. Retrieval with PLSR

Figure 5 shows the retrieval results with PLSR. With six or more wavelengths, the retrieval accuracy reached the highest and remained stable, and the accuracy in calibration and the corresponding validation was very close. For example, at 120 wavelengths, based on *R* and using the sum of the |*β*| rankings as the wavelength importance indicator, the R^2^ values were both 0.67, and the RMSE was 4.65 and 4.71 nmol/cm^2^ in calibration and validation, respectively. The PLSR retrieval based on log(1/*R*) was remarkably better than that based on *R*. However, compared to Figure 3, the PLSR retrieval was generally clearly worse than those of the GPR models. To obtain an accurate PLSR model, the retrieval based on log(1/*R*) was further examined.

Table 5 lists the retrieval results of the last seven models with SBBR. From iterations #1–7 with strategies 1 and 2, the wavelengths in the final models were selected in sequence. The retrieval results with the two final PLSR models are listed in Table 6. These two models achieved almost the same accuracy, and this accuracy was also the same as the models with ≥ 6 wavelengths in Table 5. With a lower number of wavelengths, the final PLSR model with strategy 1 was simpler than that with strategy 2. Compared to the two final GPR models in Table 4, the two final PLSR models were more complicated and less accurate. Figure 6 shows the scatterplots for the measured vs. predicted content using the two final PLSR models in validation. With the measured content < 18 nmol/cm^2^, the two models performed well, while with content greater than this level, the predictions became worse and showed an underestimation. For example, with the #1 model, at content less than and greater than this level, the *r* and RMSE were 0.94 (*p* < 0.01), 2.34 nmol/cm^2^, and 0.28 (*p* = 0.11), 4.98 nmol/cm^2^, respectively.

### 3.4. Retrieval with Other Methods

The retrieval accuracy using Red/Green-1, -2, mACI, and PROSPECT-D were low, with all R^2^ ≤ 0.81 and RMSE > 3.6 nmol/cm^2^. For ARI and mARI, the results were more accurate (Table 7). These two vegetation indices performed better than the two final PLSR models (Table 6); however, they were slightly worse than the two final GPR models (Table 4).

## 4. Discussion

### 4.1. Comparison among the Retrieval Methods

From Section 3, we can see that GPR performed the best among all the investigated methods. GPR is one of the machine learning techniques. In recent decades, machine learning has been being extensively applied in the remote sensing of vegetation biochemical content (e.g., water, chlorophyll), and can achieve higher accuracy than other methods (e.g., [40,41,42,43]). Our study also demonstrated this situation. There may be several reasons for this. First, considering the influence of other factors (e.g., chlorophyll absorption), the relationship of leaf green reflectance vs. anthocyanin content was not linear. Therefore, the linear models that directly use reflectance as independent variables and that fail to effectively reduce the influence were disturbed, causing a loss in accuracy (PLSR, Red/Green-1, -2, and mACI). However, the non-linear machine learning methods may grasp the relationship of green reflectance vs. anthocyanin content well (see also Section 4.3). Second, the machine learning methods can utilize many spectral wavelengths, and make optimization to specific datasets. Vegetation indices use only several wavelengths, and thus may omit potential useful information. Third, the main reason for good performance of ARI and mARI was that they significantly reduced the interference from chlorophyll absorption with the item −1/*R*_band_ (band is around 705 nm) [10,11].

For the radiative transfer models (PROSPECT-D here), they describe the physical interaction between leaf and light, and are simplification and generalization to this interaction. Their accuracy in prediction mainly depends on how well the process is understood and accounted for in modeling [44]. Thus, these models do not necessarily entirely adapt to some specific circumstances. Nevertheless, with the physical basis, radiative transfer models do not need calibration each time and can obtain consistent accuracy among datasets, while the empirical methods (e.g., vegetation indices and machine learning) should be calibrated first before use and the accuracy among datasets may vary greatly.

In short, machine learning methods have not been reported to retrieve leaf anthocyanin content to date. Our results demonstrated the capability of GPR, and set a precedent. It reminds us that other machine learning algorithms such as ANN may be also effective and are worth trying. This needs further study.

### 4.2. The Most Important Wavelengths Selected and Performance of the Obtained Models

As shown in Figure 7, across wavelengths, the correlation coefficient for *R* vs. anthocyanin content had the opposite trend with that for log(1/*R*) vs. the content, and both correlation coefficient series had the highest values in the green region. This range is also where anthocyanins have the greatest absorbance, as anthocyanins’ molar extinction coefficients show. After the logarithmic transformation, the correlation in the major absorption region of anthocyanins (436–650 nm) was enhanced overall. Most importantly, the highest correlation value increased from −0.69 to 0.83. This may explain the better performance of log(1/*R*)-based models than the *R-*based ones. Gitelson and Solovchenko also demonstrated the higher accuracy in anthocyanin content retrieval using absorbance (represented with –log(transmittance)) as opposed to using *R* [17].

From the wavelengths in the several last PLSR and GPR models (Table 3 and Table 5), three main spectral regions held during model building with SBBR (Figure 7). The first one was at 557–566 nm. This was around the green peak and corresponded to the strongest absorption region of anthocyanins. The second region was at 705–725 nm, which was around the red edge region. This region was the transition area from the intense absorption of chlorophylls to the strong scattering of the leaf structure. Although anthocyanins absorb weakly in this range, containing this range in the model may allow for reduction in the interference from the absorption by chlorophylls in the green region, as in ARI and mARI.

In addition, although the #2 model in Table 4 does not use a red edge region wavelength, the #5 model in Table 3 with strategy 2 uses it (705 nm). According to Table 3, when selecting 557 and 705 nm to build a GPR model, the retrieval result (R^2^/RMSE) was 0.93/2.18 nmol/cm^2^ for the calibration and 0.93/2.22 nmol/cm^2^ for the validation. This was slightly better than the #2 model in Table 4. Furthermore, all two-wavelength combinations with one wavelength around the green peak and the other around red edge region were selected to build two-wavelength GPR models. The retrieval results were illustrated in Figure 8. As the figure reveals, the higher the R^2^ value in a model, the lower the RMSE that this model achieved. The calibration and validation had nearly the same pattern, indicating little accuracy difference (the maximum: 0.008 for R^2^ and 0.01 nmol/cm^2^ for RMSE). All models had R^2^ > 0.78, and 75.03% of models in the calibration and 74.19% in the validation had R^2^ > 0.90. Clearly, the importance of the green peak and red edge region for building simple and accurate two-wavelength GPR models was manifested.

The third spectral region was at 743–755 nm. This range was in the near-infrared region and was used only by PLSR. The reflectance in this region was closely related to the leaf structure. Containing it in the retrieval may help to distinguish the effect of the leaf structure to lead to a more robust and accurate model.

These three regions have also proved essential in building anthocyanin vegetation indices [10,11]. In addition, it should be noted that the final models obtained based on the SBBR method do not assure the highest retrieval accuracy, as the #2 model in Table 4 implied (also refer to Figure 8). To build the most accurate model, the models on all wavelength combinations must be examined and compared. However, this is not usually practical and necessary. First, it is cumbersome and computationally intensive to examine all combinations. As for the TOTAL dataset, there were as many as 2^345^−1 wavelength combinations in total. Second, even if it is possible to test all combinations, the risk of overfitting must be considered, and the most accurate model is likely too complicated to interpret and transfer on other datasets. Therefore, a variable (wavelength) selection method is to find a “good” set of variables rather than the “best” set of variables [46].

### 4.3. Performance of the Linear PLSR vs. the Non-Linear GPR Methods

Many remote sensing methods for pigment content retrieval are primarily based on the specific absorption features of the pigment. For example, the NDVI (normalized difference vegetation index) utilizes the strong absorption of chlorophylls in the red region. The two anthocyanin vegetation indices, ARI and mARI (Table 1), use the strong absorption of anthocyanins in the green region. From Section 4.2, the PLSR and GPR methods captured this basis as well.

According to the application conditions of the Beer–Lambert law, at a low concentration level, the absorbance vs. the concentration of a medium has a linear relationship. However, at a high level, the molar extinction coefficient of the medium does not hold, and the absorbance vs. the concentration will deviate from the law. Therefore, a linear model may not properly characterize the whole process for the medium concentration growing from a low to a high level. This likely explains why the two final PLSR models performed well for the anthocyanin content < 18 nmol/cm^2^, while, for a higher content level, their performance became worse. Unlike the linear PLSR, GPR is a non-linear method. With this non-linear property, it may be more effective to characterize the relationship for absorbance vs. content.

Figure 9 shows the anthocyanin absorbance at 550 nm as a function of content and the prediction with the linear and GPR regression, based on this absorbance. At 550 nm, there were also other absorbers (mainly chlorophylls). In the red edge region, chlorophylls were the primary absorbers while anthocyanins absorbed weakly. log(1/*R*_550_) − log(1/*R*_708_) can be used as an estimate of the absorption of anthocyanins at 550 nm [17]. From the figure, the variation trend of the dot distribution was not linear, indicating a non-linear relationship for the anthocyanin absorbance vs. content. The linear model apparently deviated from the dot distribution, while the GPR model captured the distribution characteristic well, resulting in rather accurate results. Therefore, a simpler and more accurate retrieval model was more likely to be obtained with GPR than with PLSR (Table 4 vs. Table 6).

GPR also provides errors for the prediction, as Figure 10 shows. For clarity, only the results for the European hazel leaves were drawn. Clearly, the measured content fell within or very close to the range of predicted content ± one standard deviation, which indicated good prediction accuracy. Thus, with such an error plot, it was easy to analyze the accuracy and uncertainty in the estimation. In remote-sensing images, the errors can be calculated for each pixel. Then, a retrieval error image, such as for the leaf area index [27,38], can be produced in an area. This image can serve as a vivid visualization tool for the retrieval result analysis.

### 4.4. Applicability of this Study on the Canopy Scale and in Other Relevant Fields

This study built simple and accurate retrieval models for anthocyanins with PLSR and GPR at the leaf scale. However, to date, retrieval at the canopy scale has not been reported yet. For the next step, using an unmanned aerial vehicle equipped with a hyperspectral camera, field experiments can be undertaken on a vegetation canopy that has abundant anthocyanins (e.g., maple woods in autumn). Then, the proposed methods should be further examined at the canopy scale. In addition, our method may also help relevant researchers to build and refine retrieval models for other applications, such as in the canopy chlorophyll content [36] and leaf area index [47] estimation.

## 5. Conclusions

Leaf anthocyanins are of great significance for plants. However, studies on anthocyanin content retrieval are quite rare, and the application of advanced techniques, such as machine learning, has not been reported. In this paper, various methods were used to retrieve the leaf anthocyanin content. The partial least squares and Gaussian process regression were specifically focused to build models as simple as possible while maintaining accuracy.

The results showed that, based on log(1/*R*), using 564 and 705 nm, GPR obtained the best accuracy (R^2^/RMSE: 0.93/2.18 nmol/cm^2^ in the calibration, and 0.93/2.20 nmol/cm^2^ in the validation) of all the investigated methods. Selected wavelengths around the green peak and the red edge region (one from each) were promising to build accurate two-wavelength GPR models with R^2^ > 0.90. PLSR did not perform as well as GPR. The final PLSR model involved four wavelengths, and the results (R^2^/RMSE) were 0.87/2.88 nmol/cm^2^ in the calibration, and 0.88/2.89 nmol/cm^2^ in the validation. GPR apparently surpassed PLSR in the retrieval. The reason was likely that the relationship for the absorbance vs. content of anthocyanins does not maintain a linear relationship as the content grows higher. The linear PLSR model deviated from this relationship, while the non-linear GPR model can characterize this relationship well.

Our study provides an effective method to build simple PLSR and GPR models to retrieve the leaf anthocyanin content while maintaining the accuracy. This broadens the possible methods for the remote sensing of leaf anthocyanin content and sets a precedent for machine learning algorithms. It may be also helpful for researchers in developing and refining models for other retrieval problems.

## Figures and Tables

**Figure 1 sensors-21-03078-f001:**
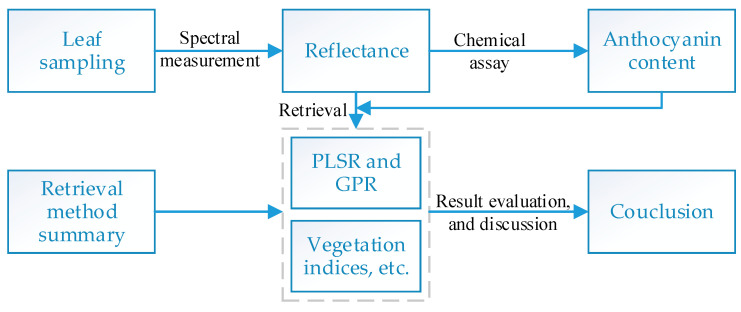
Flowchart of the research procedure.

**Figure 2 sensors-21-03078-f002:**
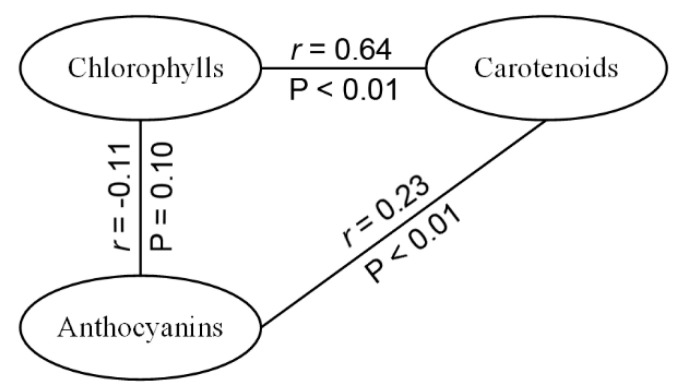
The correlation coefficient (*r*) between the pigment contents of the TOTAL dataset and the corresponding significance level (P). For *r* between anthocyanins and two other pigments, HAZEL was excluded from TOTAL.

**Figure 3 sensors-21-03078-f003:**
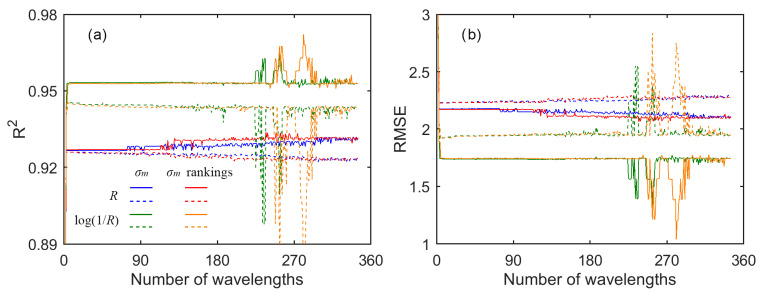
The retrieval (**a**) R^2^ and (**b**) RMSE values as a function of the number of wavelengths with GPR. The “*R*” and “log(1/*R*)”, and the “*σ_m_*” and “*σ_m_* rankings”, correspond to the data used and the wavelength importance indicator, respectively. The solid line represents the calibration, while the dashed line represents the validation.

**Figure 4 sensors-21-03078-f004:**
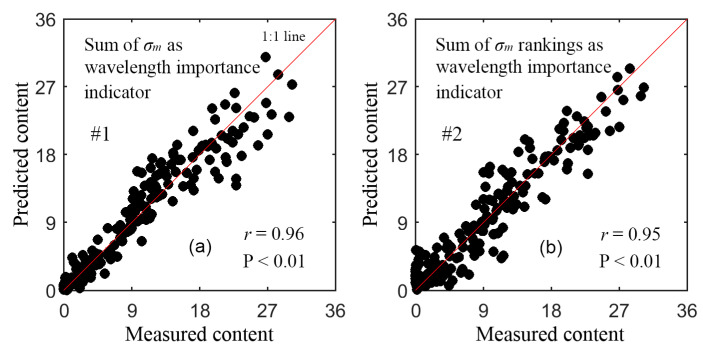
Scatterplots for the measured anthocyanin content (nmol/cm^2^) vs. predicted content with the final GPR models in Table 4. (**a**) shows the results of the #1 model, and (**b**) shows those of #2 model.

**Figure 5 sensors-21-03078-f005:**
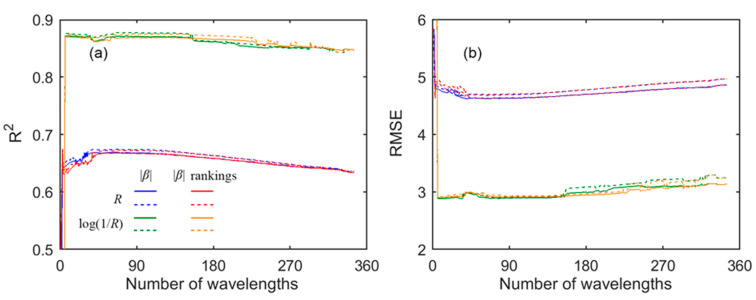
The retrieval (**a**) R^2^ and (**b**) RMSE values as a function of the number of wavelengths with PLSR. The “*R*” and “log(1/*R*)”, and the “*σ_m_*” and “*σ_m_* rankings”, correspond to the data used and the wavelength importance indicator, respectively. The solid line represents the calibration, while the dashed line represents the validation.

**Figure 6 sensors-21-03078-f006:**
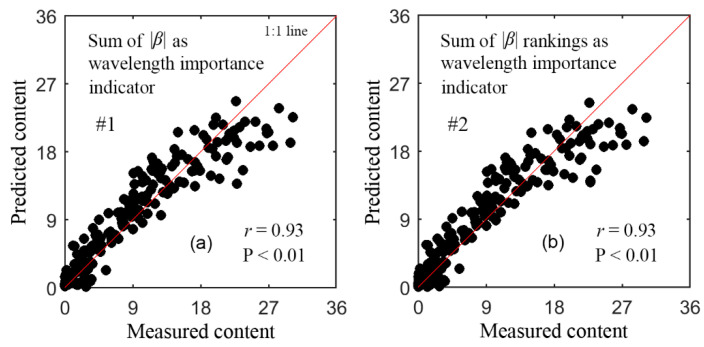
Scatterplots for the measured anthocyanin content (nmol/cm^2^) vs. predicted content with the final PLSR models in Table 6. (**a**) shows the results of the #1 model, and (**b**) shows those of #2 model.

**Figure 7 sensors-21-03078-f007:**
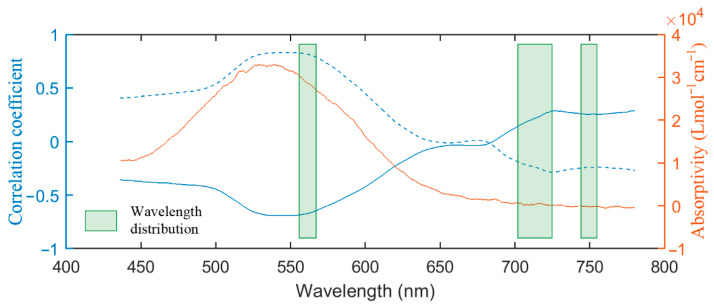
The correlation coefficient for *R* (solid line) and log (1/*R*) (dashed line) vs. the anthocyanin content (nmol/cm^2^), and the molar extinction coefficient of anthocyanins obtained from thin-layer chromatography extraction [45]. The green shaded area is the main distribution of the wavelengths of the models in Table 3 and Table 5.

**Figure 8 sensors-21-03078-f008:**
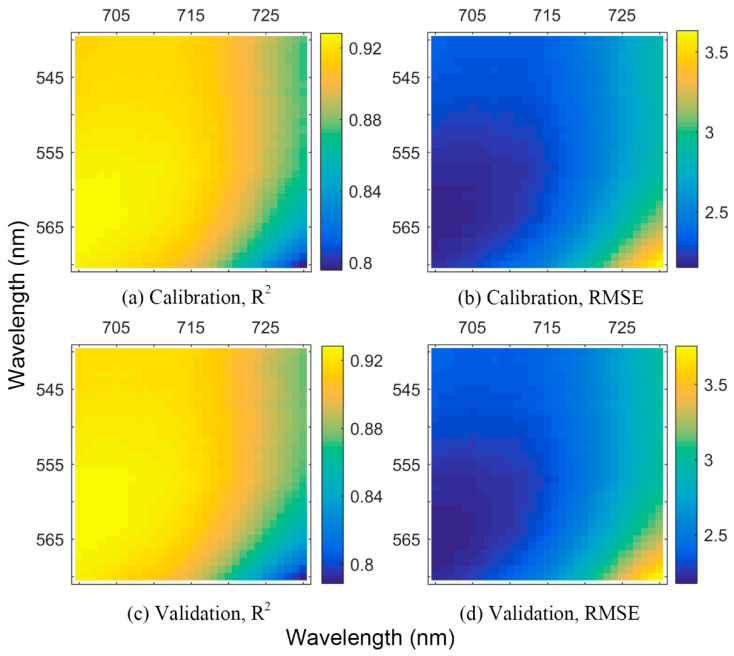
The retrieval of two-wavelength GPR models based on log(1/*R*). One wavelength was selected around the green peak (540–570 nm) and the other around the red edge region (700–730 nm). (**a**,**b**), respectively, displays the R^2^ and RMSE values in the calibration, and (**c**,**d**) displays those in the validation. The unit of the RMSE is nmol/cm^2^.

**Figure 9 sensors-21-03078-f009:**
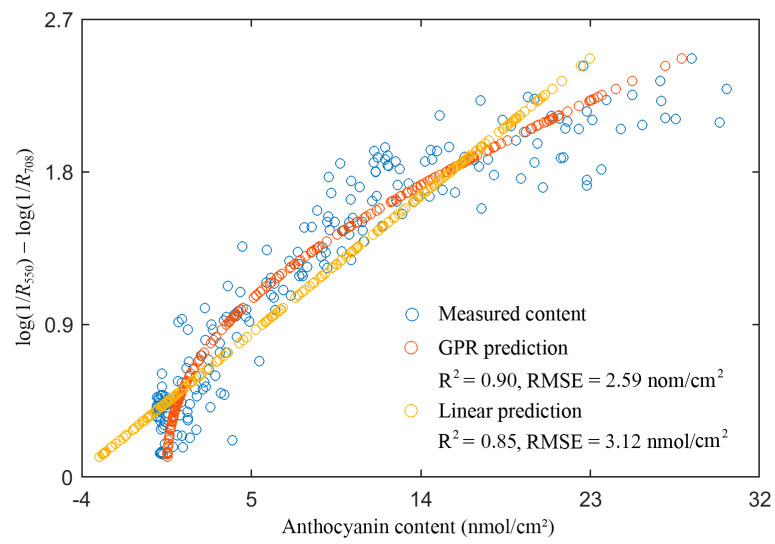
The anthocyanin content (nmol/cm^2^) prediction with a linear and a GPR model using log(1/*R*_550_) − log(1/*R*_708_) on the TOTAL dataset.

**Figure 10 sensors-21-03078-f010:**
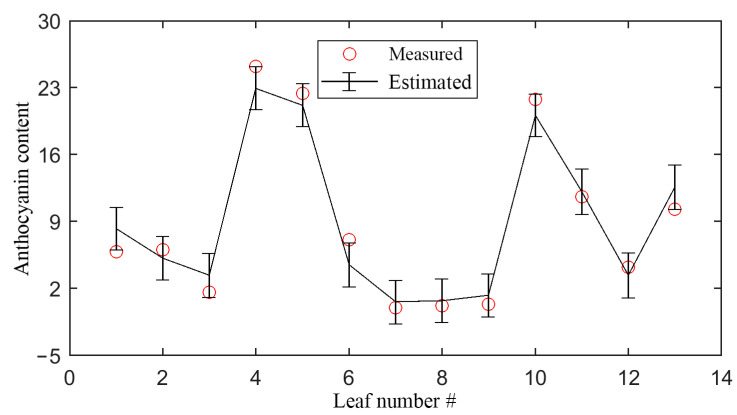
The error bar plot for the anthocyanin content (nmol/cm^2^) estimation for the 13 European hazel leaves using the #1 GPR model in Table 4. The bar represents one standard deviation, i.e., $\sqrt{\Sigma_{*}}$ in Equation (10).

**Table 1 sensors-21-03078-t001:** Other methods to retrieve the anthocyanin content.

Name	Formula	Reference
Red/Green-1	$\frac{R_{\mathrm{red}}}{R_{\mathrm{green}}}$	This study
Red/Green-2	$\frac{\sum_{600}^{699}R_{i}}{\sum_{500}^{599}R_{i}}$	[8,9]
ARI	$\frac{1}{R_{550}}-\frac{1}{R_{700}}$	[10]
mARI	$(\frac{1}{R_{530-570}}-\frac{1}{R_{690-710}})\times R_{\mathrm{NIR}}$	[11]
mACI	$\frac{R_{\mathrm{NIR}}}{R_{\mathrm{green}}}$	[15]
PROSPECT-D		[16]

**Table 2 sensors-21-03078-t002:** Description of the datasets. HAZEL did not have records of the carotenoid content, so HAZEL was excluded for the carotenoid statistics of the TOTAL dataset. The original CREEPER had 81 leaves; however, there were six leaves whose anthocyanin content was missing. Therefore, these six samples were removed. SD stands for standard deviation.

Dataset	Reference	Spectral Range (nm)	Number of Leaves	Pigment Content (nmol/cm^2^)					
				Chlorophylls		Carotenoids		Anthocyanins	
				Mean ± SD	Range	Mean ± SD	Range	Mean ± SD	Range
DOGWOOD1	[10,31,32]	436–796	23	5.13 ± 5.33	0.07–15.05	3.10 ± 2.22	0.42–7.88	8.64 ± 7.04	0.40–22.82
DOGWOOD2	[11]	400–1017	51	23.77 ± 7.58	1.53–39.81	5.39 ± 2.26	1.73–10.76	12.71 ± 8.21	1.07–30.23
HAZEL	[31,39]	400–800	13	26.37 ± 3.55	22.69–34.62	None	None	7.13 ± 4.19	0.25–13.61
MAPLE	[11,31,32]	400–780	48	7.43 ± 7.36	0.14–32.98	5.25 ± 2.37	1.82–10.40	8.75 ± 6.83	1.12–21.67
CREEPER	[31,39]	400–800	75	11.79 ± 14.92	0.09–53.76	3.13 ± 3.12	0.15–12.27	6.72 ± 8.66	0.00–26.97
TOTAL		436–780	210	13.88 ± 12.71	0.07–53.76	4.23 ± 2.85	0.15–12.27	8.88 ± 8.05	0.00–30.23

**Table 3 sensors-21-03078-t003:** GPR retrieval of the last five iterations, using the SBBR algorithm based on log(1/*R*). The underlined are the selected wavelengths for the final GPR models.

No.	R^2^		RMSE (nmol/cm^2^)		Wavelength (nm)
	Calibration	Validation	Calibration	Validation	
*Strategy 1: sum of σ_m_ as the wavelength importance indicator*					
#1	0.81	0.82	3.47	3.53	564
#2	0.87	0.88	2.89	2.93	564, 566
#3	0.93	0.93	2.18	2.23	564, 566, 705
#4	0.93	0.93	2.18	2.23	560, 564, 566, 705
#5	0.93	0.93	2.18	2.23	560, 561, 564, 566, 705
*Strategy 2: sum of σ_m_ rankings as the wavelength importance indicator*					
#1	0.84	0.84	3.25	3.32	557
#2	0.87	0.88	2.90	2.96	557, 566
#3	0.94	0.94	1.89	2.03	477, 557, 566
#4	0.94	0.94	1.89	2.03	477, 557, 564, 566
#5	0.95	0.95	1.75	1.91	477, 557, 564, 566, 705

**Table 4 sensors-21-03078-t004:** Retrieval results with the final GPR models (log(1/*R*) based).

No.	Wavelength Importance Indicator	R^2^		RMSE (nmol/cm^2^)		Wavelength (nm)
		Calibration	Validation	Calibration	Validation	
#1	Sum of *σ*_m_	0.93	0.93	2.17	2.20	564, 705
#2	Sum of *σ*_m_ rankings	0.92	0.92	2.21	2.37	477, 557

**Table 5 sensors-21-03078-t005:** PLSR retrieval of the last seven iterations, using the SBBR algorithm based on log(1/*R*). The underlined are the selected wavelengths for the final PLSR models.

No.	R^2^		RMSE (nmol/cm^2^)		Wavelength (nm)
	Calibration	Validation	Calibration	Validation	
*Strategy 1: sum of* |*β*| *as the wavelength importance indicator*					
#1	0.08	0.12	7.71	7.71	723
#2	0.09	0.12	7.67	7.74	723, 755
#3	0.09	0.12	7.67	7.74	722, 723, 755
#4	0.09	0.02	7.66	7.74	709, 722, 723, 755
#5	0.09	0.12	7.66	7.74	707, 709, 722, 723, 755
#6	0.87	0.88	2.88	2.89	566, 707, 709, 722, 723, 755
#7	0.87	0.88	2.88	2.90	566, 707, 709, 722, 723, 743, 755
*Strategy 2: sum of* |*β*| *rankings as the wavelength importance indicator*					
#1	0.08	0.12	7.69	7.70	725
#2	0.09	0.11	7.68	7.75	725, 744
#3	0.09	0.12	7.67	7.75	725, 744, 755
#4	0.09	0.12	7.67	7.75	725, 744, 754, 755
#5	0.09	0.12	7.66	7.74	707, 725, 744, 754, 755
#6	0.87	0.87	2.90	2.94	564, 707, 725, 744, 754, 755
#7	0.87	0.87	2.90	2.94	564, 707, 723, 725, 744, 754, 755

**Table 6 sensors-21-03078-t006:** The retrieval results with the final PLSR models (log(1/*R*) based).

No.	Wavelength Importance indicator	R^2^		RMSE (nmol/cm^2^)		Wavelength (nm)
		Calibration	Validation	Calibration	Validation	
#1	Sum of |*β*|	0.87	0.88	2.88	2.89	566, 709, 723, 755
#2	Sum of |*β*| rankings	0.87	0.87	2.90	2.93	564, 707, 725, 744, 755

**Table 7 sensors-21-03078-t007:** The retrieval results with ARI and mARI.

Method	R^2^		RMSE (nmol/cm^2^)	
	Calibration	Validation	Calibration	Validation
ARI	0.91	0.91	2.45	2.43
mARI	0.90	0.91	2.54	2.54

## Data Availability

The data presented in this article are available on request of the corresponding author.

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
