# Peer review of "Leaf Anthocyanin Content Retrieval with Partial Least Squares and Gaussian Process Regression from Spectral Reflectance Data"

_sensors, 2021, doi:10.3390/s21093078_

Round 1
Reviewer 1 Report
Comments are written in attached file.

Author Response
Please see the attachment. Thank you again for your time and detailed reviewing.

Reviewer 2 Report
Dear authors,
although the paper is interesting, it's topic is not in line with paper aim. Therefore, some lacks can be detected. Please, read the following detailed comments.
- Introduction section: Please, provide more details about the advantages and disadvantages of traditional techniques and remote sensing technique;
- Introduction section: Please, improve the description of RS methods. How many techniques have been developed? and compare their performance;
- Introduction section: The proposed aim is not in line wih manuscript content. Please, re-phrase it adapting it to the document (e.g. "Other retrieval methods" are not mentioned);
- Introduction section: Please, insert a short description of subsequent paper sections
- Introduction section: Which is the novelty of your reseach?
- Methodology: include an additional paragraph to explain the operative workflow;
- " The datasets" section: please, provide more information regarding the used spectroradiometer (i.e. accuracy, spectral resolution,....);
- " The datasets" section: Move figure 1 in Results (add an extra paragraph to describe datasets feattures);
- "Model calibration, validation and evaluation" section: please, re-write it since it is not clear the adopted methodology;
- Improve discussion section. Which are strengths and weakness of different methods? Compare your results with literature outcomes;
- Which are future developments of your research?
Author Response

(The authors gave the same response as above.)

Reviewer 3 Report
The subject of the article is interesting, but there are some small bugs that should be corrected:
- To improve this work's quality, I suggest that authors carefully review the text (according to the journal template) by checking the grammar and correcting the typos, grammar errors, and other shortcomings found in the document.
- I think that a thorough review of the English language is mandatory.
- Figures (3,5,7) captions are unnecessarily long and repetitive (is it useful to repeat the same description for each one of them?)
- Last but not least, I think that a summary table explaining all the symbols and acronyms used in this work would be beneficial for improving its readability and comprehensibility; therefore, I suggest including this list of symbols/acronyms in a dedicated section of this paper.
- The summary has three paragraphs, each of which represents the summary of the paper. The abstract must be written in a more objective way, with only one abstract, and must meet the norms of the newspaper (200 words maximum);
- In point 4.1, the authors write about machine learning of the artificial neural network. Please describe the learning plan. How were the training patterns prepared, what was the form of these vectors? For the reader, the presented research results are important, but more important is how the research was conducted.
Round 2
Reviewer 2 Report
Dear authors,
although the paper was strongly improved, paper quality is not still satisfying.
- the aim is not in line with paper content since
- "Specifically, PLSR and GPR were specifically investigated." Actually you reported addiotional techniques (as as spectral vegetation) too;
- "To our knowledge, it was the first time that these two techniques were applied in leaf anthocyanin content retrieval.". PLSR was apllied in several studies to retrielval Leaf Anthocyanin. Probably, you should improve the loterature review;
- The operative workflow section should be moved up. It should help the readers in following the methology. Thus, there is no sense in putting it at the end of "methodology" section;
- Results should be deeply commented. For instamce, what are the meaning of Table 2?
- Improve the english style of the manuscript
